# Molecular Identification and Prevalence of the Mite *Carpoglyphus lactis* (Acarina: Carpoglyphidae) in *Apis mellifera* in the Republic of Korea

**DOI:** 10.3390/insects15040271

**Published:** 2024-04-14

**Authors:** Thi-Thu Nguyen, Mi-Sun Yoo, Hyang-Sim Lee, So-Youn Youn, Se-Ji Lee, Su-Kyoung Seo, Jaemyung Kim, Yun-Sang Cho

**Affiliations:** 1Laboratory of Parasitic and Honeybee Diseases, Bacterial Disease Division, Department of Animal and Plant Health Research, Animal and Plant Quarantine Agency, Gimcheon 39660, Republic of Korea; nguyenthush.hua@gmail.com (T.-T.N.); msyoo99@korea.kr (M.-S.Y.); leehs76@korea.kr (H.-S.L.); syyoun@korea.kr (S.-Y.Y.); dltpwl86@korea.kr (S.-J.L.); ssk922@korea.kr (S.-K.S.); kimjm88@korea.kr (J.K.); 2Institute of Biotechnology, Vietnam Academy of Science & Technology, Ha Noi 11300, Vietnam

**Keywords:** *Apis mellifera*, *Carpoglyphus lactis*, Cytochrome *c* oxidase subunit 1 (*COI*) gene, molecular identification, Republic of Korea

## Abstract

**Simple Summary:**

Honeybees (*Apis mellifera*) are crucial for our ecosystem, but they face various threats. This study investigates the detection of a new mite species emerging in Korean honeybees, *Carpoglyphus lactis*. Using the polymerase chain reaction method to amplify the Cytochrome *c* oxidase subunit 1 gene region, we examined the presence of *C. lactis* in honeybee colonies across nine provinces of the Republic of Korea. We found *C. lactis* in eight provinces in ROK, particularly during winter. This is the first identification of *C. lactis* in Korean honeybees. Identifying new threats is essential for beekeepers and researchers. Understanding the prevalence of *C. lactis* mite species and its impact on honeybee health will help develop strategies to protect bee populations.

**Abstract:**

*Apis mellifera*, especially weak ones, are highly vulnerable to *Carpoglyphus lactis* mites, which can rapidly infest and consume stored pollen, leading to weakened colonies and potential colony collapse. This study aimed to ascertain and investigate the prevalence of this mite in honeybee colonies across nine provinces in the Republic of Korea (ROK). A total of 615 honeybee colony samples were collected from 66 apiaries during the spring and 58 apiaries during the summer of 2023. A 1242 bp segment of the Cytochrome *c* oxidase subunit 1 (*COI*) gene was amplified using the polymerase chain reaction method. The detection levels of *C. lactis* in the honeybees were compared between winter and summer. Based on the *COI* sequence analysis, the nucleotide sequence similarity of *C. lactis* mites isolated in the ROK with those from China (NC048990.1) was found to be 99.5%, and with those from the United Kingdom (KY922482.1) was 99.3%. This study is the first report of *C. lactis* in Korean apiaries. The findings of this study demonstrate a significantly higher detection rate in winter, which is 4.1 times greater than that in summer (*p* < 0.001). Furthermore, the results underscore the usefulness of molecular diagnostic techniques for detecting *C. lactis* mites.

## 1. Introduction

The honeybee (*Apis mellifera*), also known as the western honeybee, plays a vital role in the world. These industrious insects act as essential pollinators, ensuring the stability and health of ecosystems [1,2]. Beyond their ecological significance, honeybees provide a wealth of economic and health benefits through the production of honey, royal jelly, propolis, and beeswax [1]. Unfortunately, the rate of honeybee colony losses has been increasing annually in many countries worldwide [3,4,5]. According to a collaborative survey conducted by the Rural Development Administration, Animal and Plant Quarantine Agency, local authorities, and the Korean Beekeeping Association in the Republic of Korea (ROK), 416,409 beehives disappeared between October 2021 and March 2022. The health and survival of honeybee colonies, particularly during the harsh winter months, are of paramount importance. However, winter presents a multitude of challenges that threaten honeybee health throughout this extended period. Consequently, winter colony losses have become a growing concern in temperate climates [3,4,5]. This is primarily due to limited foraging opportunities as temperatures drop and flowers become scarce. As a result, bees rely heavily on pre-stored food reserves within the hive to survive winter. However, the prolonged presence of these stored food resources can also create a potential disease transmission risk within bee colonies. This is primarily due to limited foraging opportunities as temperatures drop and flowers become scarce. As a result, bees rely heavily on pre-stored food reserves within the hive to survive winter. However, the prolonged presence of these stored food resources can also create a potential disease transmission risk within bee colonies. Numerous factors influence the health of honeybee colonies, and one of the devastating factors in colony loss is parasitic infection, especially mite species [1,6,7,8,9,10,11,12]. In the ROK, significant harm to honeybee colonies is attributed to *Varroa* mite [1,11,13,14]. Research has also documented the presence of other potentially harmful mite species via morphological and/or genetic analysis, including *Tropilaelaps mercedesae*, *Tyrophagus curvipenis*, and *Acarapis* mites in the ROK [12,14]. However, this is the first report of *Carpoglyphus lactis* within the Korean honeybee population.

*Carpoglyphus*, commonly known as the cheese mite or milk mite, belongs to the genus *Carpoglyphus* within the family Carpoglyphidae of the Astigmata order [15,16,17]. This mite is primarily associated with cheese and other milk products [16]. The previous reports mainly focus on the relationship between the mite and milk products, as well as its impact on cheese quality. This species has been observed in the Holarctic, Oriental, Australian, and Indian regions [15,18,19,20,21]. Additionally, *C. lactis* mites are known to be relatively common in beehives at different positions within the observed hive. They are primarily found in stored pollen and aged beehives, potentially posing a threat to stored honey [22]. Moreover, they are present in beehive debris, in hive materials, on dead bees, and on beehive frames, serving as a source of pollen, honey, and essential bee substances stored in queen bee cells [23,24]. This ability to exploit various resources within the hive highlights their potential to disrupt and harm bee colonies. The detrimental impact of *C. lactis* becomes particularly severe during the winter. Germany and the United States highlight widespread cases of beehive destruction due to *C. lactis* mites during winter storage [20,25,26]. In Zander, Germany, 250 beehives were reportedly destroyed by *C. lactis* mites in a single winter storage area back in 1947 [27]. Similarly, research by Baker and Delfinado in 1978 documented cases in Alabama of the US where beehives were found heavily infested with *C. lactis* mites following winter storage [21]. These observations reveal a devastating impact, with *C. lactis* mites consuming stored pollen, polluting the hive with a brownish-yellow powder of mite carcasses, debris, and pollen, and ultimately destroying the combs [20,25,26]. While previous research has focused on the relationship between *C. lactis* and stored food products, information regarding its detection rate and prevalence within honeybee colonies remains limited. Given the potential role of *C. lactis* infestation for bee health, understanding its presence and prevalence is crucial for beekeepers to develop effective management strategies. Accurate identification of mite species is crucial for understanding their impact on honeybee health. Based on DNA sequencing, the Cytochrome *c* oxidase subunit I (*COI*) gene has proven to be a valuable tool in mite phylogeny studies. In this study, the molecular method was applied to detect the presence of *C. lactis* mites within honeybee colonies across ROK.

This study aimed to identify the *C. lactis* mites in honeybee colonies in the ROK in 2023. We conducted a nationwide survey encompassing nine provinces to determine the detection rate of this mite species. By establishing a baseline understanding of *C. lactis* prevalence in Korean honeybees, this research marks the initiation of investigations for future reference and encourages the development of effective management strategies against *C. lactis* in honeybee colonies.

## 2. Materials and Methods

### 2.1. Collection of Samples and Detection of Astigmatid Mites in Honeybees

Honeybee samples were collected from 9 provinces (Gangwon, Gyeonggi, Chungcheonnam, Chungcheonbuk, Gyeongsangnam, Gyeongsangbuk, Jeollanam, Jeollabuk, and Jeju) in the ROK (Figure 1). The honeybee samples were regularly collected throughout the year 2023 from non-migrating honeybee colonies. Honeybee samples were collected from the hive of each opened colony; subsequently, the comb was taken out after carefully observing the comb without the queen bee. A total of 615 honeybee colonies from 124 apiaries were sampled to investigate the presence of Astigmatid mites. The worker bees (*n* = 10~30) were collected in 50 mL Falcon tubes from colonies in each apiary. Larvae samples were collected from three random locations within the hive using forceps, checking 10 larvae at each site.

Samples were collected using sterile tools to prevent contamination and directly placed into 50 mL Falcon tubes. Each tube was labeled with a unique identifier to ensure sample traceability throughout the testing process. To preserve the integrity of the collected samples, they were promptly transferred to a temperature-controlled storage facility at −20 °C. The honeybee samples were used to assess temperature-dependent appearance patterns. Concurrently, these samples were used to evaluate the prevalence, diversity, and simultaneous occurrence of pathogens and Astigmatid mites. The samples were divided into two portions; the first was used for Astigmatid mite examination under a dissecting microscope, and the remaining one was used for total nucleic extraction in polymerase chain reaction (PCR). For beehives that tested positive for *C. lactis* mites, the combs in honeybee colonies were collected during the next sampling round for further laboratory examination. These samples were observed under a dissecting microscope to confirm the presence and morphology of this mite. As this study only involved honeybees, which are invertebrates not covered by most ethical regulations related to animal research, formal ethical approval was not required.

### 2.2. Total Nucleic Acid Extraction

The honeybee samples were homogenized using a Precellys 24 tissue homogenizer (Bertin Instruments, Montigny-le-Bretonneux, France) in three cycles of 15 s at a speed of 5000 rpm in 1 mL of phosphate-buffered saline solution. Subsequently, a mixture of 200 μL of lysis buffer and 20 μL of proteinase K solution was added to the homogenized sample, and the resulting mixture was incubated at 56 °C for 10 min. The total nucleic acids were extracted using the Maxwell^®^ RSC automated system according to the manufacturer’s instructions. Finally, 60 μL of purified total nucleic acids were employed to detect honeybee pathogens. The total nucleic acids from the honeybee and control samples were stored at −20 °C until further analysis.

### 2.3. Amplification of Astigmatid Mite-Specific Genes Using Polymerase Chain Reaction

DNA sequence-based identification has proven to be the most reliable approach for species determination. The *COI* gene has been successfully used in the species identification and phylogenetic studies of mites [28]. A specific primer pair targeting the *COI* gene of *C. lactis* mites was designed and used for PCR amplification. The primer sequence was derived from a reference *C. lactis COI* gene sequence deposited in GenBank (https://www.ncbi.nlm.nih.gov/, accessed on 18 Mar 2020) under accession number KY922482.1. This approach ensured the amplification of the desired target gene and minimized the possibility of amplifying DNA from other organisms. The primer pairs used in this experiment are described in Table 1.

The total reaction volume (20 µL), including 1 µL of DNA, 1 µL of each primer (10 pmol), and 17 µL ddH_2_O, was used by AccuPower^®^ PCR PreMix and Master Mix (Bioneer Corp., Daejeon, Republic of Korea). The PCR thermal cycling conditions were as per those by Nguyen et al. [12]: 94 °C (5 min); 5 cycles of 94 °C (20 s), 52 °C (30 s), and 68 °C (30 s); 5 cycles of 94 °C (20 s), 50 °C (30 s), and 68 °C (30 s); 30 cycles of 94 °C (20 s), 48 °C (30 s), and 68 °C (30 s); 30 cycles of 94 °C (20 s), 46 °C (30 s), and 68 °C (30 s); and a final extension step at 68 °C (5 min). The amplified products corresponding to the *COI* gene segment of *C. lactis* in the honeybee colonies were visualized using 1% agarose gel electrophoresis, with a determined size of 1242 bp. Subsequently, the PCR products were purified and sequenced by Cosmogenetech Co., Ltd. (Seoul, Republic of Korea). To verify the presence of *C. lactis* mite and assess its genetic diversity within Korean honeybees, the positive control was specifically designed by cloning the *COI* gene from a known Korean *C. lactis* mite.

### 2.4. Phylogenetic Analysis and Statistical Analysis

The sequences of the *COI* gene segments of *C. lactis* mites amplified from different regions were sequenced and compared with the *COI* gene of this mite available in the National Center for Biotechnology Information (NCBI). Subsequently, the sequences were aligned using ClustalW multiple alignment in BioEdit version 7.2.5 [29]. A species phylogenetic tree was constructed using MEGA11 based on the nucleotide sequences of the *COI* gene [30]. Simultaneously, a phylogenetic tree was built according to the calculated values obtained from Kimura’s two-parameter distance method by neighbor-joining with 1000 bootstrap replicates and the maximum-likelihood method. Three sequences of *C. lactis* mites isolated from China (NCBI accession nos.: NC_048990.1 and MN073839.1) and the UK (NCBI accession no. KY922482.1) were chosen as representative sequences.

The values were evaluated using the likelihood ratio chi-square test to measure the strength between the variables of the two groups. Statistical significance was set at *p* < 0.05.

## 3. Results

### 3.1. Molecular Identification of Carpoglyphus lactis Mites in Honeybee Colonies

Among 615 honeybee colonies sampled from 124 apiaries, PCR analysis successfully identified the presence of the *C. lactis* mite in a subset of colonies. The characteristic 1242 bp PCR product confirmed the presence of the mite (Figure 2).

Comparative analysis of the gene segment sequences of *C. lactis* mites compared with the GenBank database (blast.ncbi.nlm.nih.gov/Blast.cgi/, accessed on 03 Apr 2023) unveiled a significant level of nucleotide similarity (99.1~99.5%) between those from our study and those from China (NCBI accession No. NC_048990.1) and the UK (NCBI accession no. KY922482.1) (Appendix A). Variations in the *COI* gene sequence of *C. lactis*, involving A to G and C to T at identical positions from the UK, were identified. These mutation positions exhibited remarkable consistency among strains isolated nationwide, resembling those from the UK (Appendix A). Analysis of the *COI* gene sequence revealed a high degree of similarity (between 99.1% and 99.5%) among *C. lactis* mites isolated from honeybee colonies across ROK (Figure 3). Notably, these sequences exhibited the greatest similarity to strains isolated from China (NCBI accession nos. MN073839.1 and NC_048990.1). Interestingly, one isolate from Gyeonggi province showed a closer match to a strain from the UK (NCBI accession no. KY922482.1), suggesting some potential genetic variation within the Korean *C. lactis* mites. Our analysis of the *COI* gene sequence in honeybee samples revealed the widespread presence of *C. lactis* in honeybee colonies across nine provinces in the ROK.

### 3.2. Similarities of COI Gene Sequence of Astigmatid Mites in Honeybee Colonies

*Astigmatid* mites are characterized by their tiny size and detrimental impact on honeybee colonies. Species identification was conducted by analyzing the phylogeny, utilizing the *COI* gene sequences of *C. lactis* compared to confirm mite species found in honeybee colonies (Figure 4). Our phylogenetic analysis, based on the *COI* gene sequences belonging to two families, aligns closely with current morphological classifications. The mites found in honeybee apiaries belong to the hyporder Astigmatid; however, within this order, the *C. lactis* mite forms a distinct and separate branch from *Tyrophagus* mites. The first brand contains *C. lactis* of the family Carpodae. The remaining three species cluster together: *T. longior* from Belgium (NCBI accession no. KY986280.1); *T. putrescentiae* from China (NCBI accession nos. MH2gyphi62535.1 and MH262542.1); and *T. curvipenis* from ROK (NCBI accession no. OQ121363.1). The *COI* nucleotide sequence of *C. lactis* mites isolated from Korean honeybee colonies exhibited a high nucleotide similarity of 80.2% with *Tyrophagus* mites.

### 3.3. Prevalence of Carpoglyphus lactis Mite in Korean Honeybee Colonies

We investigated *C. lactis* mites in honeybee colonies collected during winter and summer in ROK in 2023 (Figure 5). The results indicated a significantly higher detection rate of this mite during winter compared to summer (χ^2^ = 62.713, *df* = 1, *p* < 0.001). The detection rate of *C. lactis* in 267 colonies from 66 apiaries collected during winter was 33.3%, whereas this rate during summer was only 8.1% in 348 colonies from 58 apiaries. The highest detection rate was observed in March, with a prevalence rate of 74.7% among the 75 honeybee colonies collected from 23 apiaries.

As shown in Figure 6, *C. lactis* mites were found in eight out of nine provinces. This mite was not detected in any of the surveyed apiaries in the Jeju province. The detection rates of *C. lactis* in different provinces were as follows: Gangwon at 16.7% (15/90); Gyeonggi at 26.8% (22/82); Chungcheongbuk at 26.5% (9/34); Chungcheongnam at 32.5% (13/40); Gyeongsangbuk at 7.8% (6/77); Gyeongsangnam at 32.6% (29/89); Jeollabuk at 17.9% (12/67); and Jeollanam at 21.2% (11/52). The highest infection rate was observed in the provinces of Chungcheonnam and Gyeongsangnam.

## 4. Discussion

This study investigated the molecular detection and prevalence of *C. lactis* mites in honeybee colonies in the ROK. This finding provides valuable insight into the occurrence, prevalence, and phylogenetic position of *C. lactis* mites in the ROK, a species known to increase mortality rates in honeybee colonies during prolonged winter periods.

We employed molecular techniques to identify *C. lactis* mites in honeybee colonies. To achieve this, we designed specific primers targeting the *COI* gene of *C. lactis* mites with an amplicon size of 1242 bp to minimize non-specific binding of PCR (Table 1). Successful amplification of the targeted *COI* gene segment confirmed the presence of *C. lactis* mites in the tested colonies. The subsequent sequencing and comparative analysis of these gene segments with reference sequences in the GenBank database revealed a substantial nucleotide similarity with previously isolated *C. lactis* strains from China and the UK, further supporting the accurate identification of the mite species. Analysis of the *COI* gene sequences revealed variations at specific positions in *C. lactis* mites collected across South Korean provinces (Appendix A). Compared to strains from other countries, these variations suggest a potential for adaptation to distinct environmental conditions within ROK. Further research is necessary to fully understand the functional significance of these variations and their possible impacts on *C. lactis* biology and behavior in the Korean beekeeping environment.

Honeybee mites constitute one of the major factors causing harm to honey bee colonies globally. They parasitize bees and act as disease vectors within the population. Consequently, they adversely impact the health and alter the healthy behavior of honeybees. As the life cycle of the mite occurs within the beehive, the dense population of these parasites can lead to contaminated honeybee products, reduced food resources, and compromised honeybee health. Recently, a *T. curvipenis* mite was reported in Korean honeybee apiaries [12]. *Carpoglyphus* and *Tyrophagus* mites that primarily feed on stored pollen and other organic materials have been found in beehives [22,27,31]. In this study, we detected *C. lactis* for the first time in the Korean honeybee in eight of nine provinces in the ROK. The nucleotide of the *COI* gene of *C. lactis* mite is similar to that of *Tyrophagus* mite at 80.2%. Building on prior research by Dabert et al. [32], our analysis of the *COI* gene further supports the close relationship between *C. lactis* and *Tyrophagus* mites within the suborder Astigmata. Notably, Dabert et al. [32] also found these two species to cluster together in their phylogenetic tree of Arachnid mites, constructed using the 18S rDNA gene. In 2023, Bowman reported that *C. lactis* and *Tyrophagus* mites could co-occur in UK beehives [15]. The discovery of *C. lactis* mite in this study, along with the recent report of *T. curvipenis* by Nguyen et al. (2023), suggests two potential implications regarding the presence of these mite species in Korean honeybee colonies. Firstly, this discovery may indicate a genuine increase in the prevalence of newly introduced mite species in ROK. Alternatively, these mite species may have already been present but remained undetected due to the lack of previous studies specifically targeting them. Our research provides the first report of *C. lactis* mites in Korean honeybee colonies, contributing to a more comprehensive understanding of the mite fauna associated with these colonies. The spread of diseases within bee populations infested with bee mites is increasing; hence, effective control measures are challenging. Therefore, investigating the types of bee mites and assessing their prevalence is crucial to sustaining honeybee health and preventing the propagation of possible harmful mites to the bee population.

Our prevalence analysis provided valuable insights into the occurrence and distribution of *C. lactis* mites in honeybee colonies in the ROK. A nationwide survey was conducted in 2023, encompassing all nine provinces of the ROK. Sampling times were strategically chosen to capture potential variations in mite presence throughout the year, with adjustments made based on observed temperature fluctuations. The observed temporal variations in the infection rate with a higher prevalence during winter warrant further investigation into potential drivers. Environmental factors and fluctuations in honeybee populations likely play a role. Previous studies have linked its presence to increased honeybee mortality, particularly during winter months [18,20,27]. The initial detection of *C. lactis* in overwintering honeybee colonies was first reported in Germany [27], suggesting a robust development of *C. lactis* mite at lower temperatures, particularly within the beehive, making it an ideal habitat for this mite species. This potential threat is concerning, especially considering the extended periods of food storage necessary for honeybee survival in temperate climates like ROK. Our findings on the high prevalence of *C. lactis* highlight the need for further research to elucidate its role in honeybee health and potential interactions with winter mortality.

Our survey revealed a significant presence of *C. lactis* mites across a substantial portion of the honeybee population sampled in the ROK. A striking 88.89% of the samples collected from eight out of nine provinces surveyed tested positive for *C. lactis* infestation. Interestingly, Jeju province, an island separated from the mainland, emerged as an exception. None of the samples collected from Jeju beehives showed any detectable presence of *C. lactis* mites, contrasting sharply with the widespread occurrence observed in the other provinces. In addition, identifying *C. lactis* in both honeybee and beeswax samples indicates that *C. lactis* mites may have a substantial impact on the honeybees’ health environmentally as well as individually. A more profound investigation into the life cycle of *C. lactis* mites in honeybee colonies is necessary to better understand the effects of mite infestation on honeybee populations and overall hive health. This knowledge will inform timely strategies and protective measures for the sustainability of beekeeping in the ROK. Until now, research on the detection and prevalence of *C. lactis* mites in honeybee colonies, as well as the associated potential risks to honeybee health, remains limited.

While our study successfully employed *COI* gene sequencing to identify *C. lactis* mites in South Korean honeybee colonies, some limitations warrant consideration for future investigations. The morphology of *C. lactis* was not observed in our study. Identifying *C. lactis* mites was not our initial objective, leading to delayed sampling for morphological confirmation during the late winter season. Winter sampling focused solely on bees and excluded mite examination in comb or beehive, thus precluding direct morphological confirmation of *C. lactis*. Summer sampling across different bee stages and materials also yielded no detections, possibly due to low prevalence in summer or dead mite presence. Furthermore, the morphology of mites within the Astigmatid order is relatively similar, which may lead to confusion during the observation. Further investigation using alternative methods is ongoing. For a more definitive confirmation of *C. lactis* presence, a complementary approach that combines genetic analysis (utilizing genes like *COI* and *ITS*) with traditional morphological examination is recommended. The detected *C. lactis* mites using PCR solely confirm their presence and do not necessarily reflect active infection. The potential impact of *C. lactis* on honeybee health and honey production remains an area requiring further investigation. While traditionally not considered a primary pest, its high prevalence and potential for competition for food resources, nest debris generation, and possible involvement in disease transmission at high infection rates warrant further study. These are some limitations; however, our study provides valuable initial insights into the presence of *C. lactis* in Korean apiaries. Further research addressing these limitations will be crucial for understanding its potential impact on honeybee health and developing effective management strategies.

## 5. Conclusions

This study provides the first report of *C. lactis* mites in honeybee colonies within the ROK. Employing molecular detection techniques, we revealed a widespread distribution of *C. lactis* across eight out of nine provinces surveyed. Furthermore, our results demonstrated a seasonal variation in prevalence, with higher detection rates observed during winter. Phylogenetic analysis confirmed the distinct identity of *C. lactis* within the suborder Astigmata and its near relationship with *Tyrophagus* mites. These findings highlight the potential threat posed by *C. lactis* to Korean honeybee health and emphasize the need for further research. Future investigations should focus on elucidating the impact of *C. lactis* infestation on bee colony health and productivity, particularly during winter.

## Figures and Tables

**Figure 1 insects-15-00271-f001:**
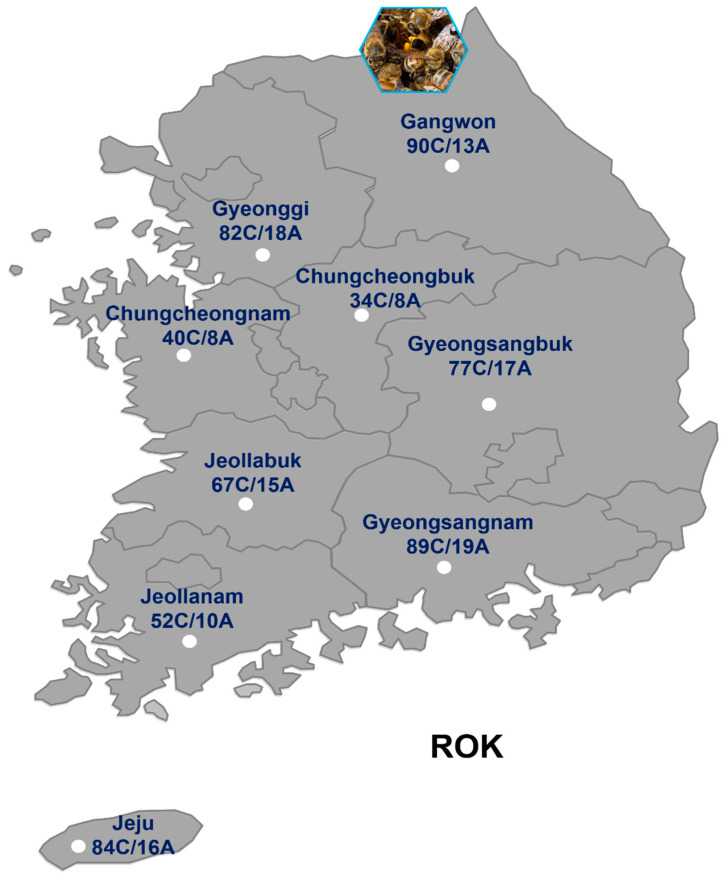
Honeybee colonies collected from 124 Korean apiaries in 9 provinces of the ROK in 2023. The number includes adult bees and larvae. C: colony; A: Apiary.

**Figure 2 insects-15-00271-f002:**
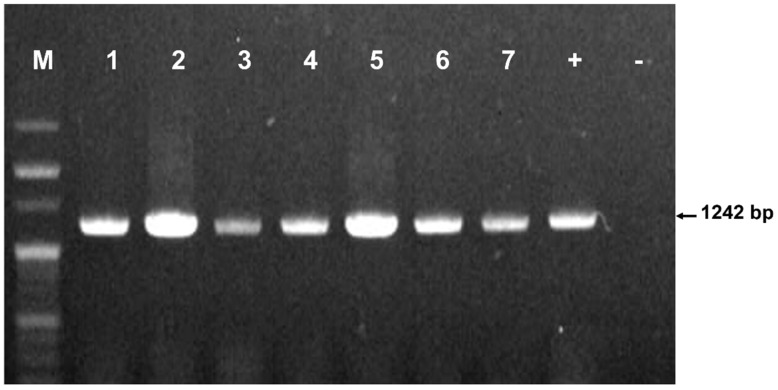
Amplification of *COI* regions from *Carpoglyphus lactis* mites in honeybee samples. PCR products were confirmed on 1% agarose gel. Lane M is a 100 bp DNA ladder (Enzymomics Co., Ltd., Daejeon, Republic of Korea); lanes 1–7 showcase PCR products of the honeybee samples; lane “+” indicates a positive control; lane “–” indicates a negative control without a DNA template.

**Figure 3 insects-15-00271-f003:**
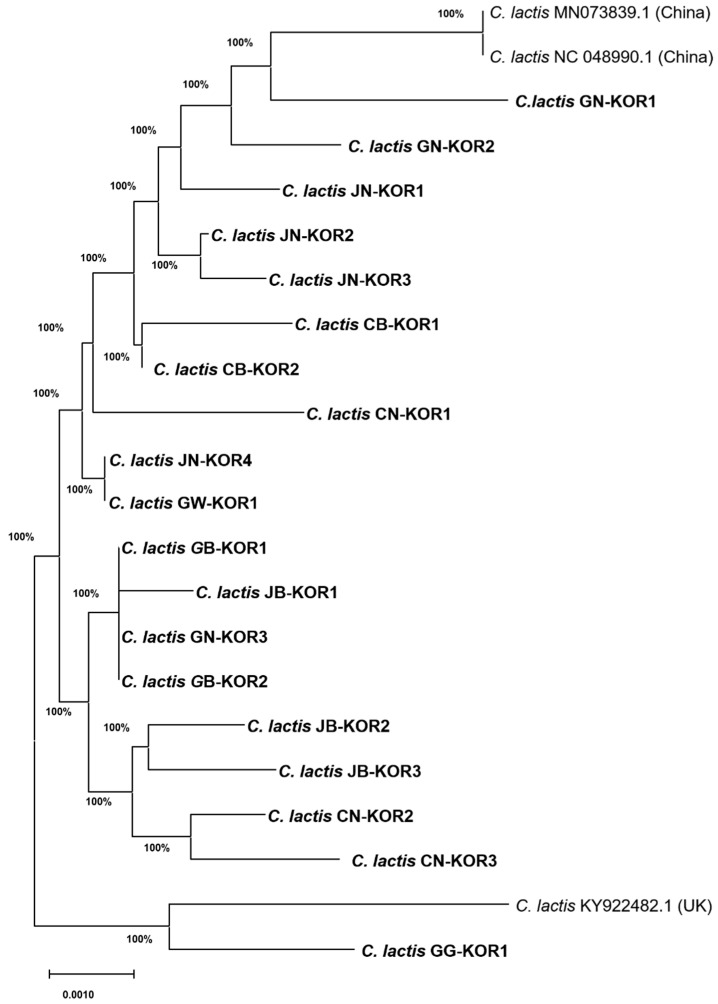
Neighbor-joining phylogenetic tree of *Carpoglyphus lactis* mite based on sequences of *COI* gene (1137 bp). *C. lactis* mite was isolated from each province of the ROK.

**Figure 4 insects-15-00271-f004:**
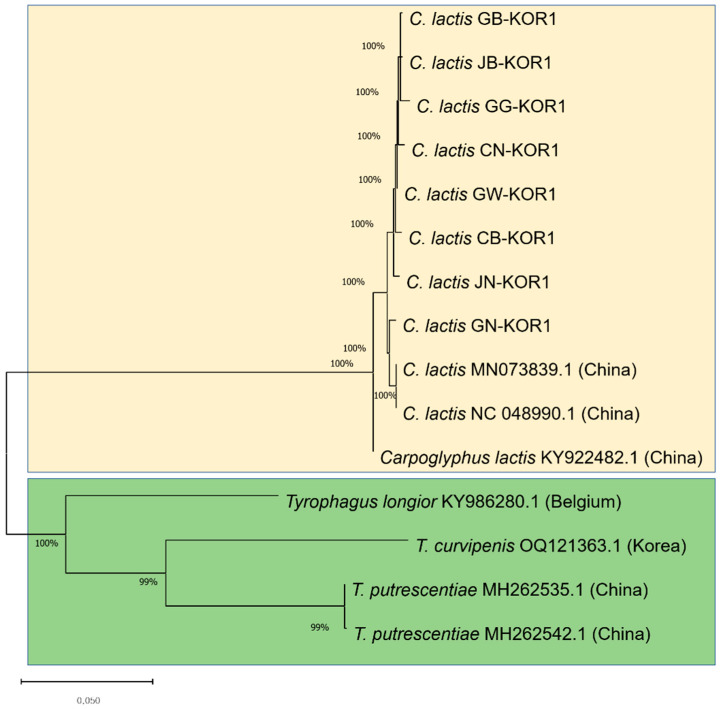
Phylogenetic maximum-likelihood tree based on sequences of *COI* gene of mites. The species name, NCBI accession numbers, and origin of the accession are given for each sequence. *Carpoglyphus* mites are highlighted in gold, *Tyrophagus* mites are highlighted in green. *Carpoglyphus lactis* (*C. lactis*), *Tyrophagus curvipenis* (*T. curvipenis*), and *Tyrophagus putrescentiae* (*T. putrescentiae*).

**Figure 5 insects-15-00271-f005:**
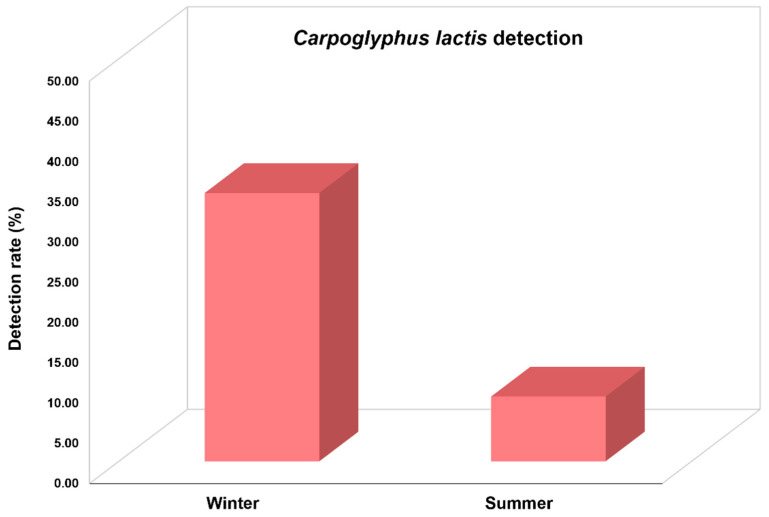
*Carpoglyphus lactis* detection rate during the winter and summer of 2023 in Korean honeybee colonies.

**Figure 6 insects-15-00271-f006:**
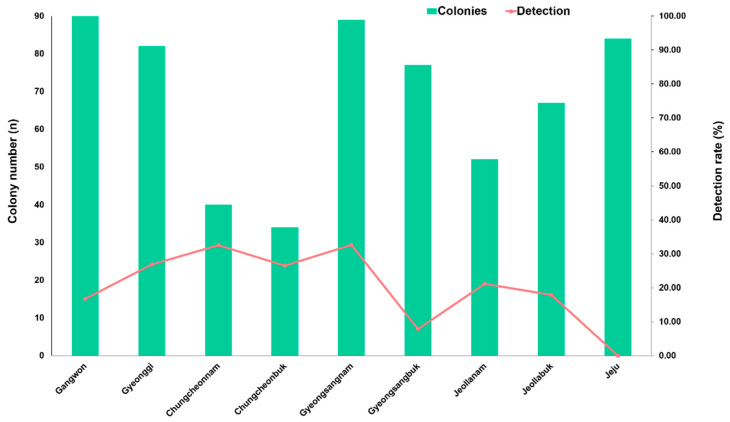
Detection rate of *Carpoglyphus lactis* mites in honeybee colonies from 9 provinces in the ROK.

**Table 1 insects-15-00271-t001:** Primers used in this study.

Name of Primers	Sequences (5’→ 3’)	Amplicon Size (bp)	Note	References
*COI*–For	GTTTTGGGATATCTCTCATAC	377	Used for Astigmatid mite detection	[28]
*COI*–Rev	GAGCAACAACATAARAAGTATC			
CL-For	CTTGAATTTGTAGAATGGA	1242	Used for *C. lactis* mite detection	This study
CL-Rev	CTAATCGAGGTGTCCGAGGT			

## Data Availability

All data are provided in this manuscript.

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
