# Peer review of "Molecular Identification and Prevalence of the Mite Carpoglyphus lactis (Acarina: Carpoglyphidae) in Apis mellifera in the Republic of Korea"

_insects, 2024, doi:10.3390/insects15040271_

Round 1

Reviewer 1 Report

Comments and Suggestions for Authors

The research yielded valuable insights into the prevalence of the mite Carpoglyphus lactis within honeybee colonies across different provinces in the Republic of Korea. By employing the polymerase chain reaction method, specifically amplifying the Cytochrome c oxidase subunit 1 (COI) gene, the study bolstered its credibility and improved mite detection accuracy. Comparing nucleotide sequence similarities with mites from diverse regions such as China and the United Kingdom reinforced the study's conclusions and provided a broader understanding of C. lactis distribution. Notably, the discovery of a significantly higher detection rate during winter compared to summer underscores seasonal variations in mite infestation levels, which could inform beekeepers in implementing tailored management strategies.

While the study acknowledged potential repercussions of mite infestation on honeybee colonies, it is worth considering that the significance of C. lactis to honeybees may have been overstated. To date, C. lactis has never been regarded as a pest of honeybees in any part of the world. 

Suggestions:

The authors’ statement about the significance of C. lactis should be backed by robust evidence.

It would be good if the authors could acknowledge differing viewpoints on the significance of C. lactis and seek feedback from colleagues or experts in the field through peer review.

Comments on the Quality of English Language

The manuscript is generally well-written. The language flows smoothly, and the ideas are communicated clearly. 

Reviewer 2 Report

Comments and Suggestions for Authors

Dear Authors,

Excellent job. This information lays a sound foundation for beekeeping in ROK.

I have indicated corrections and comments in the text for your consideration, after which it can be published.

Comments on the Quality of English Language

Good

Reviewer 3 Report

Comments and Suggestions for Authors

I appreciate the opportunity to evaluate this manuscript.  

In my opinion, minor revision is needed in order that the manuscript can be published. I will address these issues in a point-by-point section after the general comments written bellow.

This article deals with Molecular identification and prevalence of the mite Carpoglyphus lactis (Acarina: Carpoglyphidae) in Apis mellifera in the Republic of Korea.

In my opinion, it brings contribution in the field of beekeeping science, having in mind the expansion potential of the investigated mite and not so many conducted investigations on this topic.

A very strong point are the newly designed specific primers targeting the COI gene of C. lactis mites.

Methodology is written in a very good manner, however figures are presented in relatively low-resolution quality and if possible, they should be replaced.

Conclusions address the main issues of the investigation and are consistent with the evidence and arguments presented in the manuscript.

The references are properly cited and written in a correct style following the guidelines. In the supplementary S1 figure legend C. lactis should be italic (C. lactis).

Lines 25-26: You did not sample whole colonies, but the colonies were sampled?

Line 39: plays

Lines 86-87: Larvae samples were collected…

Lines 97-98: What do you mean by “positive honeybee colonies”. Please rephrase.

Lines 156-157: As I see it, it should state: “lanes 1-7 showcase PCR products of the honeybee samples indicating positive finding; lane “+” indicates a positive control; lane “-“ indicates a negative control without a DNA template”.

Line 160: …from our study…

Lines 181-184: This sentence misses a finishing, please rephrase.

Lines 184-185: Which nucleotides of C. lactis exhibited similarity? Please rephrase.

Comments on the Quality of English Language

Quality of English Language is very good.
